# Using the AraBERT Model for Customer Satisfaction Classification of Telecom Sectors in Saudi Arabia

**DOI:** 10.3390/brainsci13010147

**Published:** 2023-01-14

**Authors:** Sulaiman Aftan, Habib Shah

**Affiliations:** 1Department of Computer Science, Texas Tech University, Lubbock, TX 79709, USA; 2Department of Computer Science, College of Computer Science, King Khalid University, Abha 62529, Saudi Arabia

**Keywords:** BERT, AraBERT, deep learning, customer satisfaction classification

## Abstract

Customer satisfaction and loyalty are essential for every business. Feedback prediction and social media classification are crucial and play a key role in accurately identifying customer satisfaction. This paper presents sentiment analysis-based customer feedback prediction based on Twitter Arabic datasets of telecommunications companies in Saudi Arabia. The human brain, which contains billions of neurons, provides feedback based on the current and past experience provided by the services and other related stakeholders. Artificial Intelligent (AI) based methods, parallel to human brain processing methods such as Deep Learning (DL) algorithms, are famous for classifying and analyzing such datasets. Comparing the Arabic Dataset to English, it is pretty challenging for typical methods to outperform in the classification or prediction tasks. Therefore, the Arabic Bidirectional Encoder Representations from Transformers (AraBERT) model was used and analyzed with various parameters such as activation functions and topologies and simulated customer satisfaction prediction takes using Arabic Twitter datasets. The prediction results were compared with two famous DL algorithms: Convolutional Neural Network (CNN) and Recurrent Neural Network (RNN). Results show that these methods have been successfully applied and obtained highly accurate classification results. AraBERT achieved the best prediction accuracy among the three ML methods, especially with Mobily and STC datasets.

## 1. Introduction

Customers play an essential role in the business of various small, medium, and large enterprises (SMEs). These businesses or industries can reach the top or bottom based on customer relations, loyalty, trust, support, feedback, opinions, surveys, and other comments, individually or combined. It is very significant to understand customers’ needs and comfort levels in commercial and individual-based industries, i.e., customer satisfaction during or after service utilization. Researchers have used different customer feedback acquisition and prediction methods, such as social platforms, electronic surveys, calls, emails, online mobile applications, and websites [1]. Based on sharing customer feedback, comments, advice, suggestions, recommendations, and views, the quality and quantity of services can be improved and extended [2].

Telecommunication is one of the crucial global fields that can play a significant role in each sector, such as business, defense, investment, production, and individual. The fast, reliable, secure, and accurate service can increase the service quality of the corresponding communication companies. Therefore, the prediction of customer feedback is essential to the country’s development. Various computer science, theoretical and mathematical, and statistical techniques have been proposed and simulated to accurately predict customer satisfaction so that service quality can be improved effectively according to customer needs and expectations [3,4]. In the Saudi Arabian region, customer feedback is taken very seriously in companies’ government and private sectors. Different departments have been established to supervise and resolve customer complaints in various sectors using different strategies and channels, which has built an extremely competitive telecom market [5]. The Saudi telecom industry is rapidly changing in technological developments, service delivery, competitive landscape, and telecommunications services expansion in the non-traditional telecom services sector. These include managed infrastructure, data center/colocation, and cloud services. Saudi Telecom Company (STC, Saudi Arabia), Integrated Telecom Company (ITC, Saudi Arabia), Saudi Mobily Company (Etisalat, Saudi Arabia), Zain, Virgin, and Go Telecom are the most famous [6]. Saudi Arabia nations are among the most populous nations in the Gulf Cooperation Council (GCC) region, where most of the population comprises young people. These young nations believe in and utilize advanced technology for education, research, business, production, and other sectors. The high-speed 5G network and COVID-19 have increased the future impact of uncertain business situations [7]. The STC, Mobily, and Zain won the 5G awards in 2021. Also, according to the source, around 11 million people are using the Twitter platform on smartphones and computers, while the percentage of users is increasing rapidly due to the population and interest [8].

These three companies are very serious about taking customer feedback and comments on various channels such as Twitter, Facebook, and websites. Suggestions, words, issues resolution, and company stock value prediction are significant in classifying customer loyalty. Previously, various techniques have been used to predict customer feedback on social media platforms [9]. Twitter in Saudi Arabia is also widely used as a source of information to predict the financial market, movements in stock markets, and others [1]. Various researchers have utilized Arabic Twitter data to forecast the outcomes of the corresponding company and provide the best analysis to customers through the emerging sentiment analysis approaches based on advancements in Natural Language Processing (NLP) and text analytics techniques to identify and evaluate the opinions of users expressed in their tweets [10].

According to the Global Competitiveness Report of the World Economic Forum, people in Saudi Arabia are included in the top 36 competitive nations out of 140 countries. Therefore, their business market, production, customer relationship, quality and honesty, and demand are among the top attractive business models in the Middle East and an open market [11]. Apart from computer science and engineering-based problems, the various novel ML and DL techniques have played an essential role in the world economy, business, customer relations, satisfaction, prediction of the trend of customers over the industry concerning time and quality, and quantity [12,13,14]. However, it is difficult to predict accurate values due to the challenges of analyzing the customer views and understanding the real meaning and classification of this feedback.

Physiological researchers have simulated the various neural network models extracted from the human brain processing systems to create intelligent machines that can perform tasks that usually require human intelligence, such as visual perception, speech recognition, decision-making, and language translation [15]. Machine learning techniques are often used in conjunction with neural networks, and both are inspired by how the brain processes information and learns from experience [15,16]. Overall, the study of the brain and how it works has significantly influenced the development of AI, and the structure and function of the brain inspire many of the techniques used in AI.

Apart from others, the most widely used technique is Sentiment analysis in various product domains using the customer online feedback dataset to identify the positive and negative suggestions, complaints, and others about the product, company, and others. In computer science, classification such as speeches, videos, customer feedback, images, and text are the most common and complex tasks in ML and other typical approaches [17,18]. Like NLP, which can analyze the various types of data for multiple tasks such as sentiment analysis, cognitive assistant, spam filtering, lexical (structure) analysis, parsing, semantic analysis, discourse integration, pragmatic analysis, detecting fake news, detection of false income, and various real-time types of language translation [19,20,21]. According to [22], different ML techniques have been used and simulations to predict telecommunication companies’ customer satisfaction based on Arabic tweets.

Generally, various methods based on DL, such as Convolution Neural Networks (CNN) [23], Recurrent Neural Networks (RNN) [24], Hierarchical Attention Networks (HAN) [25], Support Vector Machine (SVM) [26], Residual Learning with Simplified CNN Extractor [27], distant, subjective supervision [28], adaptive recursive neural network [29], Random Forest (RF), Decision Tree (DT) [30], Bidirectional Long Short-Term Memory (Bi-LSTM), a hybrid of CNN and Bi-LSTM, Naive Bayes (NB) [31], Emotion Tokens, BiGRU-CNN model [32], Improved Negation Handling, and other effective intelligent methods for classification of the Turkish, Chinese, Thai, Covid, business, and medical-based Twitter datasets for sentiment analysis [33,34]. For Arabic language tweets, in the datasets analysis for various tasks such as classification or prediction, the researchers have used such Deep Attentional Bidirectional LSTM, Chi-Square and K-Nearest Neighbor, Convolutional Neural Networks, Narrow Convolutional Neural Networks (NCNN), CNN and RNN, Bidirectional LSTM, SVM, KNN, Decision Trees, NB, and others for Arabic Sentiment Analysis using the Twitter dataset for solving different tasks [35,36].

Predicting the Saudi stock exchange market, including STC datasets, VM, KNN, and Naive Bayes methods, has been simulated successfully with higher accuracy of 97.10 and 95.71% using SVM Precision and Recall phases, respectively [37]. It has been found that DL outperforms typical ML in various performance measures [38]. However, the classification/prediction accuracy can be increased using the emerging DL-based model such as BERT, which will be discussed next.

DL methods have also shown promising results in classification, prediction, and sentiment analysis in the last decades [39]. Especially in sentiment analysis, DL methods outperform the bag-of-words approach in feature generation. A few ways with limited datasets were found to get accurate simulation results in Saudi Telecommunications Companies, which are not enough for future trends, classifications, and service improvement based on customer tweets. For this purpose, deep learning-based methods such as CNN, RNN, and the recent AraBERT techniques have been proposed to predict the accurate results of these critical companies named STC, Zain, and Mobily. The AraBERT model has been chosen because it has been trained on a large corpus of Arabic text and can be used for various NLP tasks such as language translation, text classification, and question answering. It can also be fine-tuned for specific tasks, such as sentiment analysis or named entity recognition [40].

The rest of the paper is organized into five sections, namely: related works that will summarize the previous work in the corresponding area; the three Deep Learning methods (CNN, RNN, and the recent AraBERT) with a short intro and parameters; proposed simulation methodology; simulation results and discussion; and conclusion to finish.

## 2. Deep Learning Methods

Sentiment analysis (SA) is one of the complex tasks of computationally identifying and categorizing the opinion of various parties expressed in different text formats. It has various contributions in multiple areas, such as forecasting market movements, quality prediction, and improvements based on sentiment in various platforms such as blogs, news, social media posts, comments, and ratings [41]. Furthermore, the complexities increase when these comments are in the form of local language, feelings, or emotions.

Based on sentiment analysis, the identifications of various customer satisfaction and dissatisfactions can be easily obtained with multiple classes and lead to recommender systems for other customers. For large global businesses, the number of customers is increasing rapidly, along with their assessments, comments, and suggestions. The conventional approaches cannot manage and identify the future risk of such significant stakeholders’ views; therefore, computer tools are needed for such complex analyses. Various computational and mathematical tools have been proposed to simulate the identifications of accurate feedback and forecasting values with these social and non-social sentiments. These methods are Naïve Bayes (NB) classifier, Long Short-Term Memory (LSTM), CNN, and other Deep Learning techniques that have established new success histories for solving various complex computer science-related problems [42,43,44,45,46]. Some of the famous types of ML algorithms have been mentioned in Figure 1.

The automated and parallel way of human processing methodologies of DL has increased the effectiveness of these methods and the motivation of researchers from various backgrounds. Many new, improved, and hybrid DL algorithms have been introduced, used, and published in high-quality journals with an outstanding performance from typical methods. Also, the researchers used various DL techniques to indicate the relationship between multiple companies and their customers based on the feedback, quality, comment, and surveys in different areas. Along with, text classification analysis in NLP these three methods have gotten more attention than others based on highly accurate results, which have played an essential role in various domains such as business and customer relations, social impacts on future trends, etc. [19,47,48]. The process includes a pretreatment, feature extraction, selection of sentiment classification algorithm, and sentiment classification performance measures in the mentioned sentiment classification. The three classification algorithms, CNN, RNN, and BERT, have been explained in the following sections.

### 2.1. Convolutional Neural Network

Convolutional Neural Network (CNN), which was initially developed in the Neural Network Image Processing Community (NNIPC), is a famous kind of Feed-forward Neural Network (FNN) with a profound structure and has shown outstanding simulation results in various tasks, particularly in NLP tasks, such as sentence analysis of various languages in different applications. Multiple types of CNN, including the typical model, have been an important focus of research as they can be applied to complex problems involving time-varying patterns [49,50,51]. The standard CNN involves two operations, which feature extractors, convolution and pooling; the obtained output is then associated with the following [23]. CNN can interpret spatial data through the convolution layers (CL). A CL has various filters or kernels, which it learns to extract specific features from the corresponding dataset. The kernel is a 2D window that has sided over the convolution operation’s input data. We use temporal convolution in our experiments to analyze sequential data like tweets. The typical CNN architecture is used to simulate the Arabic tweets datasets of STC, Zain, and Mobily for classification purposes, as given in the following Figure 2.

The different layers of CNN are: Convolutional Layer (the first layer used to extract the features from the given dataset using mathematical operation), the Pooling Layer (using pooling operations to decrease the computational costs), the Fully Connected Layer (consists of the weights and biases values and neurons for connection to others layers), Dropout (to overcome the overfitting problem, some neurons can be dropped), and Activation Functions (to produce the output in the desired template with the suitable functions), which are considered in the typical model.

Previously, CNN and its various versions have been successfully used to classify and predict different business models based on customer feedback and reviews. In 2021, the CNN model was used, along with RNN and RoBERT, to predict the rating given to the products [14]. The simulation results obtained by CNN for Tweet sentiment analysis, which contains 377,616 geotagged tweets, achieved the highest accuracy of 66.0%, based on the one text feature. In contrast, the highest accuracy of 78.0% is achieved using a combination of text and count of nearby location categories features. This accuracy has been increased to 74 and 78 % by using the CNN using a pre-trained 6B GloVe model. It is noted that the CNN using the pre-trained 27B GloVe model achieved the highest accuracy, 83.9% and 94%, respectively, for the same datasets [52]. The hybrid version of CNN and LSTM, called the ensemble model, also successfully used for the classification of the Arabic Twitter dataset by [53], achieves an F1-score of 64.46%, which outperforms the state-of-the-art deep learning model’s F1-score of 53.6%, which is higher than CNN and LSTM versions.

Another hybrid version of the Convolutional Neural Network and Differential Evolution Algorithm called DE-CNN is proposed and simulated on various Arabic twitter datasets, achieving high accuracy and being less time-consuming than the state-of-the-art algorithms [54]. Using five types of the Twitter dataset, Deep CNN has been simulated successfully and obtained outstanding results for sentiment classification [55]. Using several techniques to classify churn prediction in the telecommunication industry, the CNN algorithm showed higher precision with a value of 97.78% [47]. In many famous significant airlines worldwide, CNN outperformed in analyzing tweets extracted based on customers’ experiences [12].

On the other hand, a study used deep learning technology to evaluate the torsional strength of Reinforced Concrete (RC) beams. The data-driven model is based on a 2D convolutional neural network (CNN) and uses information such as the beam width, concrete compressive strength, etc., in the model inputs. An improved bird swarm algorithm (IBSA) was used to optimize the hyperparameters of CNN, which was then tested using a dataset of 268 groups of lab tests of RC beams. The results showed that the proposed 2D CNN outperforms other machine learning models, building codes and empirical formulas in evaluation metrics [56]. Diagnosing surface cracks of concrete structures is essential for assessing the safety of a structure. However, traditional methods for doing this are often time-consuming and not very accurate. This paper proposes a new way of identifying surface conditions of concrete structures using a computer vision-based automated method. This method uses different convolutional neural networks (CNNs), which are pre-trained and can be used to make predictions. A modified Dempster-Shafer algorithm combines different CNN results to get more accurate predictions and creates a more reliable result. This method is checked with different types and noise levels and has been tested in real-world scenarios, which shows that it is potentially very accurate in identifying surface cracks of concrete structures [57].

### 2.2. Recurrent Neural Networks

The Recurrent Neural Network (RNN), developed in 1990, is a practical simulation resulting in modeling, classification, and other complex tasks in science, engineering, medical, and industrial areas. Aside from the others, RNN is successfully used in NLP applications such as text classifications and analysis based on various data acquisitions. The First order RNN uses context units to store the output of the state neurons from the computation of previous time steps [58]. One of the famous RNN architectures among NLP researchers is Long-Short Term Memory (LSTM). Figure 3 is proposed to simulate the three mentioned datasets for the classification task. The LSTM is simulated here because it is outstanding in solving problems such as tagging, sequence-to-sequence predictions, language modeling, and other complex computer science and engineering issues [59].

Each LSTM has a memory cell, input gate (i_t_
$i_{t}$), output gate (O_t_
$O_{t}$), a forget gate (f_t_
$f_{t}$), and hidden state (h_t_$\;h_{t}$) in a classical recurrent neural network (RNN), where the typical equation is:(1)$$S_{t}=tanh\left(U_{x\;t}+W_{s{\;}_{t-1}}\right)$$
(2)Y^=SoftmaxVs t

This paper will simulate the RNN with LSTM, expressed in the following step-by-step equations from the input to output gate: below.

Equation of Input Gate of RNN (LSTM):(3)$$i_{t}=\sigma \left(W_{i}\;h_{{\;}_{t-1}}+U_{i}x_{t}+b_{i}\right)$$
(4)$${\tilde{C}}_{t}=tanh\left(Wh_{{\;}_{t-1}}+Ux_{t}+b\right)$$

Equation of Forget Gate of RNN (LSTM):(5)$$f_{t}=\sigma \left(W_{f}\;h_{{\;}_{t-1}}+U_{f}x_{t}+b_{f}\right)$$

Equation of Memory State of RNN (LSTM):(6)$$C_{t}=f_{t}*C_{t-1}+i_{t}*\;{\tilde{C}}_{t}$$

Equation of Output Gate of RNN (LSTM):(7)$$O_{t}=\sigma \left(W_{o}\;h_{{\;}_{t-1}}+U_{o}x_{t}+b_{o}\right)$$

Using seven models on 12 different forecasting problems, RNN (LSTM) obtained the most accurate forecasts [60]. In Customer Lifetime Value (CLV), RNN was used for churn prediction and performed better than other algorithms [61]. Moreover, the recurrent neural network results improved the accuracy of the financial and social network Stock Twits [62]. The various RNN models have been applied to client loyalty number (CLN) applications. Furthermore, we have obtained state-of-the-art results in customer prediction, recency, frequency, and monetary (RFM) variables. Here, the RNN will be used to classify the Twitter dataset of Saudi telecommunications companies along with the CNN and BERT models.

### 2.3. BERT Model

The latest model, BERT, adopts the structure of a transformer, which includes multiple encoded layers, proposed in 2018, and has shown its advantages on many NLP tasks, like inference and semantic understanding, classification, etc. [63]. BERT is a bidirectional DL Model that looks to text from the two sides, left and right, rather than one side. It is a pre-trained model for various tasks, feature-based, fine-tuning, etc. The performance of BERT depends on the nature of the datasets, tasks, and Encoder (self-attention and Decoder) transfer parts.

Transformer structure is fast to process words simultaneously, and the context of words is better learned as they can understand the context from both directions simultaneously. The transformer includes two major components: the encoder and decoder layers. The Google AI developer, back in 2018, took advantage of transformer components by working on the encoder layers (self-attention and feed-forward neural network (FFNN), and proposed BERT: Bidirectional Encoder Representations from Transformers. BERT is intended to jointly condition both the left and right context in all layers to pre-train deep bidirectional representations from an unlabeled text [63].

Training BERT should be in two phases. The first phase is pre-training, where the model understands language and context, and the second phase is fine-tuning, where the model learns the language and how to solve the problems. BERT can solve many issues, such as Neural Machine Translation, Question Answering, Sentiment Analysis, Text summarization, etc. Also, BERT achieved state-of-the-art outcomes in more than 11 NLP tasks [63].

The first phase is pre-training unsupervised datasets simultaneously by two techniques. The first is masking out some percentage of the words in the input and then conditioning each word bidirectionally to predict the masked words with Masked Language Modeling (MLM). In this stage, they used WordPiece embedding, which is subword tokenization that enables the model to process the unknown words by decomposing them into known subwords [63].

As shown in Figure 4, every embedding token has a particular classification token at the beginning of every sentence [CLS] and uses [SEP] to separate them. Also, to help the model differentiate among the different sentences, they add a learned embedding indicating sentence A or sentence B is added to each token, which is segment embedding. In this process, *E* has been marked as input embedding at the first hidden vector of [CLS] token as C ∈ ℝ ^*H*^. Furthermore, at the end of the final hidden subscriber, the *i*th input token can be *T*_𝒾_ ∈ ℝ *^H^*. [2]

The second technique is understanding the relationship between sentences, which is essential in this model on Question Answering (QA) and Natural Language Inference (NLI), by applying the Next Sentence Prediction (NSP) classification task to predict whether sentence B immediately follows sentence A. For each pre-training example, the sentences A and B are chosen randomly from the corpus, with 50% of the time B being the sentence that follows A (labeled as IsNext) and 50% being a random sentence (labeled as NotNext) [63].

The capability of the transformer’s self-attention mechanism to simulate many downstream tasks, whether they entail single texts or pairs of texts, by simply swapping out the appropriate inputs and outputs makes fine-tuning simple. As encoding a concatenated text pair with self-attention effectively involves bidirectional cross-attention between two sentences, BERT uses this method to combine both stages. Feed the inputs and outputs particular to each task into BERT and fine-tune all the parameters end-to-end [63].

The second phase is fine-tuning the model, where the pre-trained parameters are used to initialize the BERT model and labeled data from the downstream tasks is used to fine-tune each parameter. Figure 5 represents the overall pre-training and fine-tuning procedures for BERT. In both pre-training and fine-tuning, the same architectures are used as given.

By pre-training the model, we have to minimize the loss; word vectors *T*_𝒾_ have the same size and are generated simultaneously. So, we have to pass these word vectors into a fully connected layered output with the same number of neurons equal to the number of tokens in the vocabulary and apply a SoftMax activation. In this way, the word vector is converted to distribution, and the actual label of this distribution would be one hot encoded vector for the exact word. So, we compare these two distributions and then train the network using the cross-entropy loss. On the other hand, the [MASK] token does not appear during the fine-tuning to prevent a conflict between pre-training and fine-tuning the model [63].

Despite being initialized with the same pre-trained parameters, each downstream task has fine-tuned parameters. For example, during the fine-tuning of classification, a weights layer *W* add where *W*
∈ℝK×H K is the number of labels. Performance is essential in the BERT model to achieve higher accuracies, so they presented two sizes of the model: BERT_BASE,_ which is “(L = 12, H = 768, A = 12, Total Parameters = 110 M) and BERT_LARGE_ (L = 24, H = 1024, A = 16, Total Parameters = 340 M)”, where L is the number of layers, H is the hidden size, and A is the number of self-attention heads [63].

#### AraBERT Model

The latest effective model, successfully used for various BERT tasks, has been improved and extended by researchers with multiple parameters and methods [3]. AraBERT is a pre-trained language model for Arabic language processing tasks. Some potential merits of using AraBERT have been trained on a large dataset of Arabic text, which means it has a strong understanding of the language and can perform tasks such as language translation, text classification, and sentiment analysis with high accuracy. Because AraBERT has already been pre-trained on a large dataset, it can be fine-tuned for specific tasks with relatively little additional training data, reducing the time and resources required to develop and deploy natural language processing models. AraBERT is specifically designed for Arabic language processing, which makes it a valuable tool for tasks that involve Arabic text. AraBERT is a pre-trained model that can be easily integrated into existing natural language processing pipelines, making it easy to use and adapt for various tasks. The ability to fine-tune pre-trained models such as AraBERT allows for transfer learning, where knowledge learned from one task can be applied to another related task, further improving performance and reducing the training data [40,64,65]. Here, the same Saudi telecommunications datasets with the standard preprocessing method, as given in the next section, have been used by the AraBERT model. Also, the AraBERT model was configured with the same topology as the typical BERT-Model, which has 12 encoder blocks, 768 hidden dimensions, 12 attention heads, 512 maximum sequence lengths, and ~110 M parameters.

Figure 6 shows the general AraBERT model using the Twitter dataset for classification purposes to know the fundamental values of customer satisfaction. The standard AraBERT model will be used to simulate the mentioned telecommunication companies’ Twitter datasets for accurate classification and CNN and RNN DL models, as given in Figure 7.

## 3. Research Methodology

The AraBERT model, successfully used in many natural and artificial-based applications with outstanding performance, especially in NLP, has never been used to predict or classify the telecommunication sector’s customer satisfaction in the Saudi Arabian region. Therefore, the AraBERT model will predict Saudi telecommunication companies’ (government and private) customer feedback quality, production, and planning. For the actual characteristics, feedback and company profile, the following models will be used for analysis with AraBERT. The RNN with Long Short-Term Memory (LSTMs) and Convolutional Neural Networks (CNN) algorithms are used as DL algorithms. The collected dataset will be preprocessed, trained, and tested with various topologies to predict customer satisfaction in these companies. The results of these DL models with different executing criteria will be measured through relevant performance metrics like accuracy, training, testing errors, etc. The preprocessed steps will be explained in Section 3.1.

In the steps trained on the LSTM model, by embedding each word in a tweet as a vector to generate word vectors embedding with 300 dimensions per word, we split the datasets into 80% training and 20% testing data. From that, we fed this model embedding words using a 64-dimensional hidden state, applied a fractional of 0.1 dropout rate over the batch of sequences, then fed the model another LSTM layer with a 64-dimensional hidden state that returns one hidden state and a single unit dense layer was applied followed by Sigmoid activation. Also, the learning rate by Adam optimizer is (lr = 10^−3^) with 10 epochs, and the Keras-TensorFlow library is implemented. On the CNN model, afterword embedding and splitting the datasets, the model is fed with several layers of convolutional layer (Conv1D with filters of 32 and kernel size of 8), and a pooling layer (MaxPooling1D with a pool size of 2) and a single unit dense layer was applied, followed by Sigmoid and ReLU activations. Also, this was with the same learning rate and epochs as the LSTM model.

In our approach, we used AraBERT based on the BERT model. There are six versions of the AraBERT model. We used AraBERTv02-large with a size of 1.38 G MB, 371 million parameters, no pre-segmentation needed, trained on 200 million sentences with a data size of 77 GB, and 8.6 billion words. Also, it contains 12 transformer blocks, 768 hidden sizes, and a self-attention of 12. In the preprocessing stage, we cleaned the data, as shown in Section 3.1. Then we pre-trained and fine-tuned each dataset spritely and merged all the datasets into one dataset to come up with the result in Section 3.3. We applied randomly splitting of 80–20% for train-test datasets. We applied the experiment settings: maximum length = 128, patch size = 16, epochs = 2, adam epsilon = 10^−8^, learning rate = 2 × 10^−5^, and GELU activation function along with Transformer and Scikit-Learn libraries.

### 3.1. Twitter Dataset

The raw dataset for this study comprised customer reviews on Saudi communications companies named AraCust, which contain: STC, Zain, and Mobily, which consisted of 20,000 observations; STC had 7590, Mobily 6450, and Zain 5950 tweets in mixed Arabic and English languages with two output positive and negative ranking [66]. The data-processing is a crucial step applied to any collected raw data before embedding it with a sentiment extraction approach with higher quality data to obtain highly accurate simulation results. The stages include stemming, cleaning datasets by removing irrelevant data, blank space, etc., and tokenization. Furthermore, all this raw data was converted into numeric conversion so that the DL algorithms could understand and deal with it. Tokenization breaks the text stream into words, phrases, symbols, or other meaningful words, and a text-to-sequences tokenizer has been used. The following steps were used to clean the data for accurate simulation classification results. The preprocessing steps for all datasets have been mentioned in the following table.

Figure 7 shows the big picture of this research, where the Twitter datasets have been simulated after preprocessing (as mentioned in Table 1) by CNN, RNN, and AraBERT for the given task.

### 3.2. Performance Evaluation Metrics

We evaluate the performance of models through the classification report that presents the main evaluation metrics of a classification-based machine learning model. It shows the model’s accuracy, recall, precision, F1 score, and support. Accuracy is the ratio of the total number of correctly predicted examples and the number of examples in the test set. The precision is the ratio of genuinely positive classifieds and the sum of true and false positive examples by the test set’s model. The recall talks about the accurate prediction of positive examples. Moreover, the F1 score is the weighted harmonic mean of precision and recall of the model.

For accurate classification analysis of the companies’ datasets, the typical performance measure matrices were used for all three methods: AraBERT, CNN, and RNN. A classification report is generated after every classification process. Mathematical models of the classification accuracy, Precision, Recall, and F-measure are given in Equations (8)–(11).
(8)$$\mathrm{Accuracy}=\frac{TP+TN\;}{TP+FP+FN+TN}$$
where TP—true positive; TN—true negative; FP—false positive; FN—false negative
(9)$$\mathrm{Precision}=\frac{TP}{TP+FP}$$

Recall—the proportion of actual positive cases that are correctly identified (Equation (10))
(10)$$\mathrm{Recall}=\frac{TP}{TP+FN}$$
where TP—true positive; FN—false negative

The F1-score is mathematically defined by Equation (11)
(11)$$F1-\mathrm{score}=2\times \frac{Precision\times Recall}{Precision+Recall}$$

### 3.3. Simulation Results on Twitter Sentiment Analysis

The three models, as mentioned above, have been simulated with different topologies and parameters for the classification purpose using three datasets: STC, Zain, and Mobily. COLAB has successfully simulated each method with the python platform. Each method’s training and testing simulation results have been evaluated based on the training and testing error and accuracy.

Table 2 shows training error results measured during the training experiments on the RNN, CNN, and AraBERT models. The CNN model has the lowest training error for all the training data sets, i.e., Zain, Mobily, and STC. While AraBERT has the highest training errors for these data sets. The AraBERT is not converging in training experiments due to its more comprehensive scope of classifying emojis and Arabic text; thus, it has significant training errors. This property makes it capable of generalizing for testing/validation data sets. In this way, it avoids overfitting.

Figure 8 shows the training accuracies obtained by the RNN, CNN, and AraBERT models. The training accuracies of RNN and CNN are higher than that of the AraBERT, leading the RNN and CNN to over-fit and thus may lead to significant errors for unknown datasets (e.g., validation datasets). The simulation results of the training experiments are further presented in the form of a confusion matrix for the three models mentioned earlier (i.e., CNN, RNN, and AraBERT).

Table 3 tabulates the confusion matrix for the CNN on the STC dataset. It precisely classifies the negative examples while giving an 8% error in the case of positive examples.

Using the CNN for the classification purpose, the results in the confusion matrix (which represents the summary of prediction results on a classification problem), obtained from the STC, ZAIN, and Mobily datasets have been presented in Table 3, Table 4 and Table 5, respectively. The average best result obtained by the CNN on the STC dataset where the F1 values are very close to 1, significantly, the weight reached 0.99, which shows that the CNN model has achieved the state of art results from other datasets (Mobily and Zain).

Furthermore, the RNN model has been successfully simulated to obtain the results in the confusion matrix of the Twitter dataset of STC, ZAIN, and Mobily companies for classification tasks. The confusion matrix values after the simulation have been presented in Table 6, Table 7 and Table 8. The results shown in Table 6 and Table 8 show that the RNN has achieved outstanding results on the STC and Zain dataset, where the weighted average reached 0.99.

The proposed AraBERT model has been simulated with the three types of Twitter datasets to evaluate the classification results in the confusion matrix. The results presented in Figure 9, Figure 10 and Figure 11 show that AraBERT also achieved the state of the results on all datasets, where the results are more accurate and stable than CNN and RNN models.

CNN, RNN, and AraBERT obtained the above simulation results on various datasets with different simulation structures and parameters.

Table 9 shows validation loss results measured after each validation experiment on the validation datasets. RNN has the lowest validation loss for the STC dataset in the validation experiments. While for Zain and Mobily, the RNN validation loss is higher than that of the AraBERT model. On average, AraBERT has the lowest validation loss for all three datasets.

Table 10 shows validation accuracy results measured after each validation experiment on the datasets. The results tabulated in Table 10 clearly show that the AraBERT has the highest average validation accuracy compared to the CNN and RNN for all the datasets.

## 4. Conclusions

This research has used Arabic Sentiment Analysis to measure customer satisfaction in Saudi Arabia Telecom Companies based on tweets. This customer satisfaction has been tested and evaluated by three methods: CNN, RNN, and AraBERT, which is based on various performance metrics, including the confusion matrix. It was found that the AraBERT model, which we simulated for the first time for the given dataset, obtained highly accurate and stable simulation results compared to other models that monitor a customer’s satisfaction on social media. The highly accurate results can be successfully used to predict customer satisfaction, providing companies with the best knowledge and early warnings.

## 5. Future Work

The authors would like to extend their current research model to multiple applications such as NLP analysis, time series dataset prediction, and classifications of various datasets using AraBERT and other DL methods. The AraBERT will be improved based on hybridization with bio-inspired and typical methods and will be simulated in a numerical time-series dataset. Also, the typical model of AraBERT will be improved based on the psychological features of customer feedback, like motivations such as the desire for convenience, quality, or value [67]. Customer feedback can reveal a person’s values, such as their priorities or beliefs about what is essential in a product or service [68], in addition to customers’ attitudes toward a company, brand, or product, including their level of loyalty or favorability [69].

## Figures and Tables

**Figure 1 brainsci-13-00147-f001:**
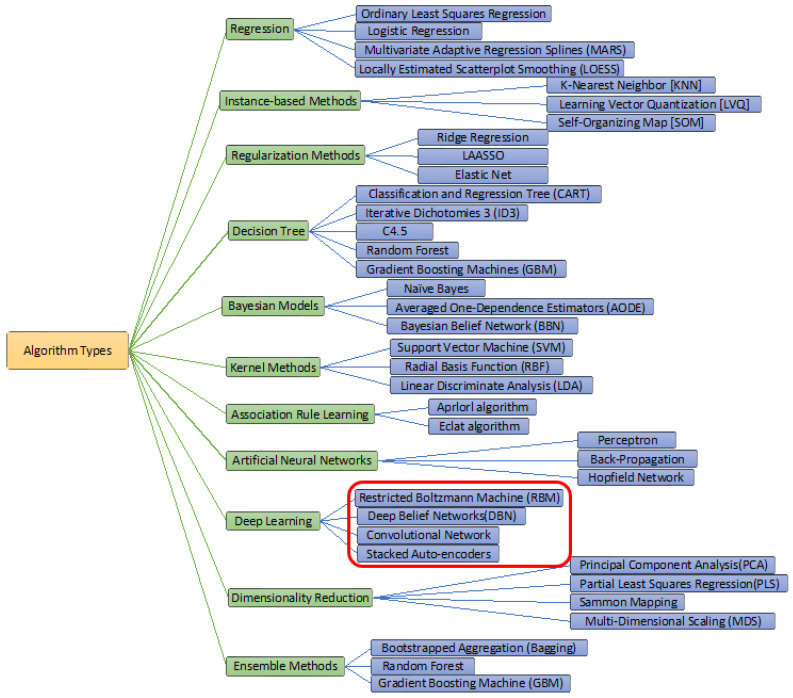
The various types of ML algorithms.

**Figure 2 brainsci-13-00147-f002:**
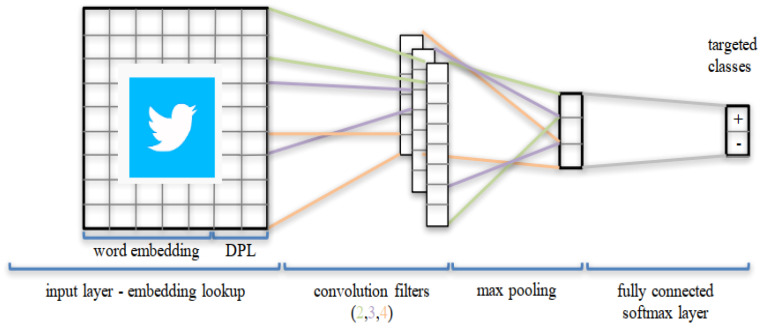
Typical CNN architecture adopted for the Twitter dataset classification purpose.

**Figure 3 brainsci-13-00147-f003:**
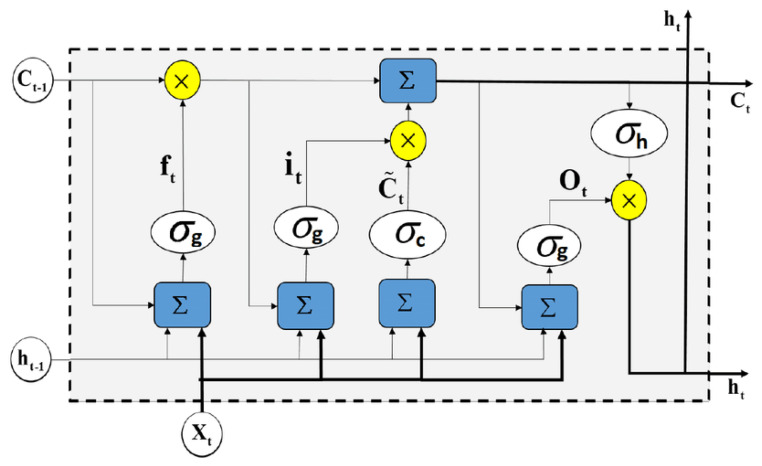
LSTM architecture cell.

**Figure 4 brainsci-13-00147-f004:**
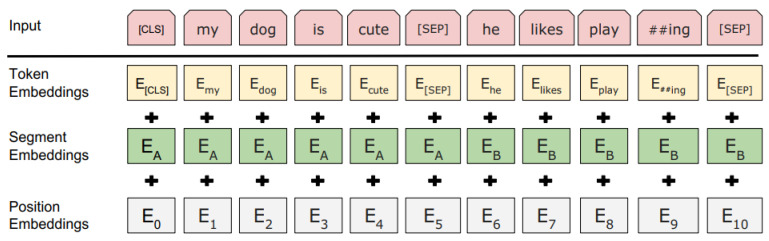
BERT input representation. The input embeddings are the sum of the token embeddings, the segmentation embeddings, and the position embeddings.

**Figure 5 brainsci-13-00147-f005:**
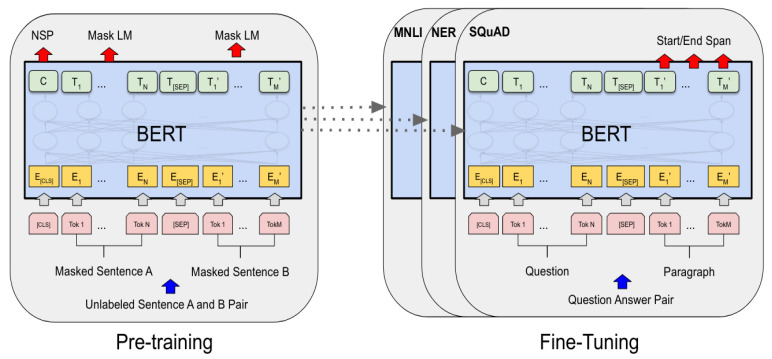
The general pre-training and fine-tuning procedures for BERT.

**Figure 6 brainsci-13-00147-f006:**
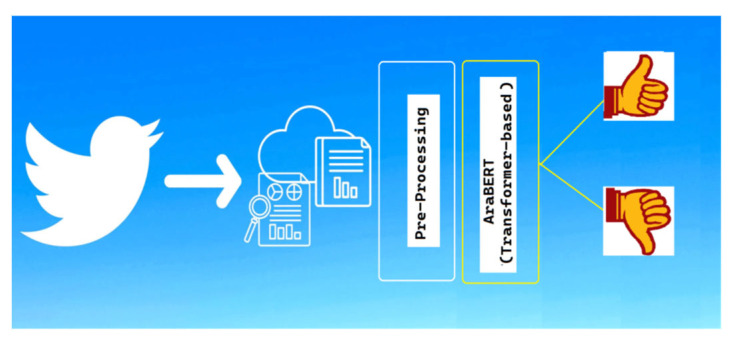
The proposed AraBERT overview.

**Figure 7 brainsci-13-00147-f007:**
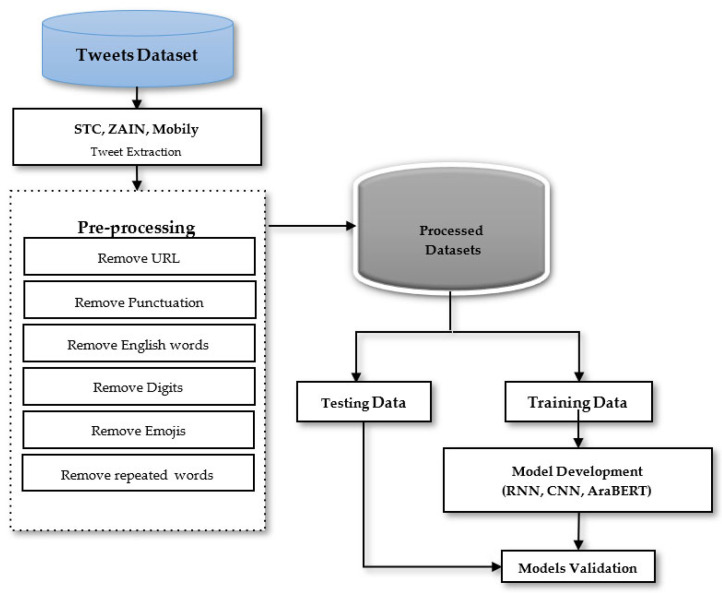
The proposed research methodology steps for classification tasks.

**Figure 8 brainsci-13-00147-f008:**
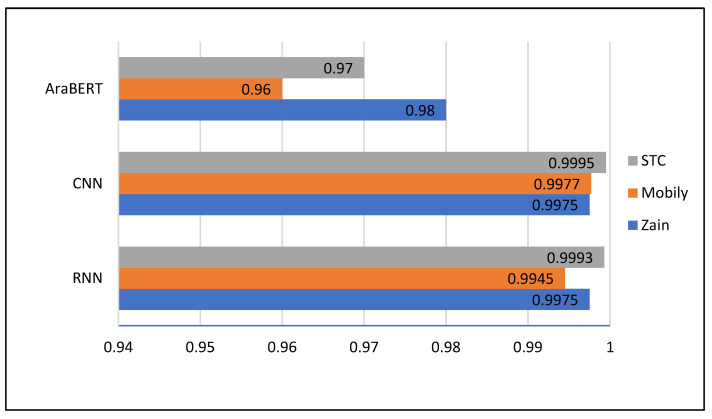
Training accuracies obtained by RNN, CNN, and AraBERT.

**Figure 9 brainsci-13-00147-f009:**
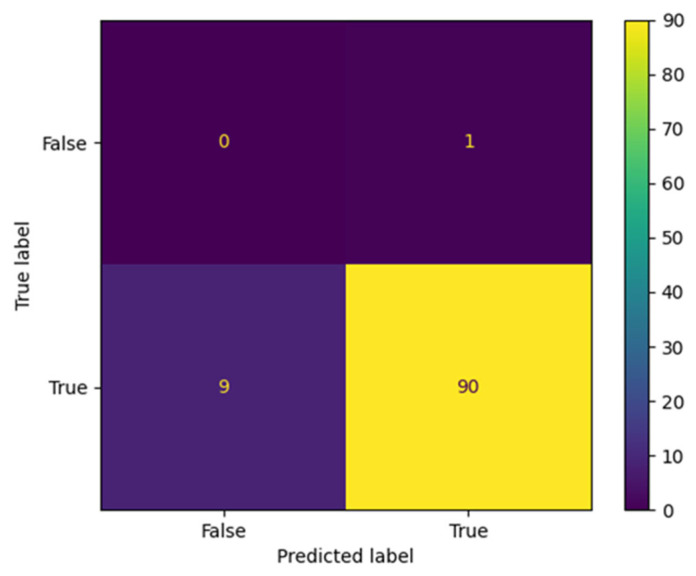
Confusion matrix obtained by AraBERT on the STC dataset.

**Figure 10 brainsci-13-00147-f010:**
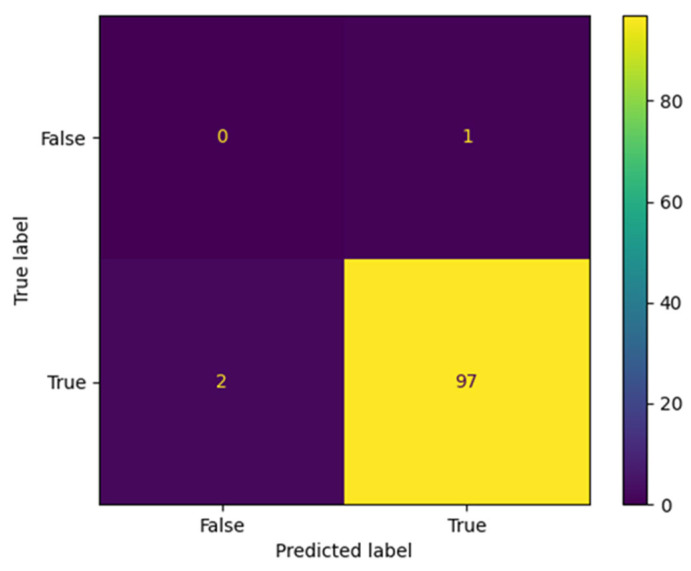
Confusion matrix obtained by AraBERT on the Zain dataset.

**Figure 11 brainsci-13-00147-f011:**
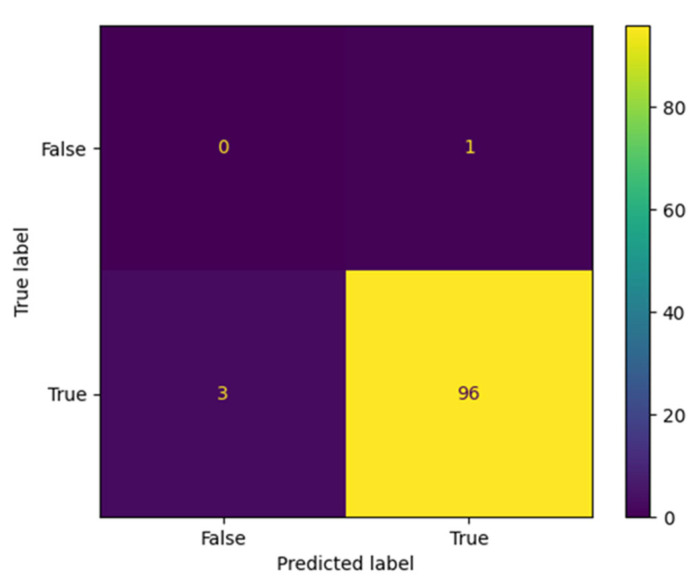
Confusion matrix obtained by AraBERT on the Mobily dataset.

**Table 1 brainsci-13-00147-t001:** Pre-processing steps of the datasets.

Preprocessing Steps		Example before Processing	Example after Processing	Translate
Step 1	Removing Emojis	@STCcare شكرا يالسمي ♥	@STCcare يالسمي شكرا	Thanks my friend
Step 2	Removing English Words	STCcare يعطيك العاافيه وشكراً لك	يعطيك العاافيه وشكراً لك	Bless you and thank you
Step 3	Removing English Symbols	@ الله يعينا ياخوك طحنا بعصابه	الله يعينا ياخوك طحنا بعصابه	May Allah help us brother We fell into this gang
Step 4	Removing Mobile No/ID/numbers	@Mobily ****05008 لاوجلا مقر نا ةلكشملا 4436**** ةيوهلا مقرو اهنا عوبسا لبق ةلكشم يدنع تناك لاير ابيرقت يدنع نم تبحس 100	رقم الجوال ورقم الهوية المشكلة ان كانت عندي مشكلة قبل اسبوع انها سحبت من عندي تقريبا ريال	Mobile number and identity number The problem is that I had a problem a week ago because it withdrew from me almost 100 riyals
Step 5	Removing stopping words	مشكور…	مشكور	Thanks
Step 6	Removing website link	متى ياموبايلي؟ https://t.co/34jDalwW4o (accessed on 10 January 2017)	متى ياموبايلي	When Mobile
Step 7	Removing repeated Arabic words	الله، شركة، اشكركم		God, a company, thank you

**Table 2 brainsci-13-00147-t002:** Training Error Results obtained by CNN, RNN and AraBERT.

Model\Dataset	Zain	Mobily	STC
RNN	0.0060	0.0229	0.0039
CNN	0.0058	0.0066	0.0032
AraBERT	0.02	0.04	0.03

**Table 3 brainsci-13-00147-t003:** Confusion matrix obtained by CNN on STC dataset.

CNN_STC	Precision	Recall	F1-Score
Negative	1.00	0.99	1.00
Positive	0.92	1.00	0.96
Accuracy	-	-	0.99
Macro average	0.96	1.00	0.98
Weighted average	0.99	0.99	0.99

**Table 4 brainsci-13-00147-t004:** Confusion Matrix obtained by CNN on the Zain dataset.

CNN_ZAIN	Precision	Recall	F1-Score
Negative	1.00	0.96	0.98
Positive	0.00	0.00	0.00
Accuracy	-	-	0.96
Macro average	0.50	0.48	0.49
Weighted average	1.00	0.96	0.98

**Table 5 brainsci-13-00147-t005:** Confusion Matrix obtained by CNN on the Mobily dataset.

CNN_Mobily	Precision	Recall	F1-Score
Negative	0.61	0.99	0.75
Positive	0.99	0.48	0.65
Accuracy	-	-	0.71
Macro average	0.80	0.74	0.70
Weighted average	0.82	0.71	0.70

**Table 6 brainsci-13-00147-t006:** Confusion matrix obtained by RNN on the STC dataset.

RNN_STC	Precision	Recall	F1-Score
Negative	1.00	0.99	1.00
Positive	0.92	1.00	0.96
Accuracy	-	-	0.99
Macro average	0.96	1.00	0.98
Weighted average	0.99	0.99	0.99

**Table 7 brainsci-13-00147-t007:** Confusion matrix obtained by RNN on the Mobily dataset.

RNN_Mobily	Precision	Recall	F1-Score
Negative	0.61	1.00	0.76
Positive	1.00	0.48	0.65
Accuracy	-	-	0.71
Macro average	0.80	0.74	0.70
Weighted average	0.82	0.71	0.70

**Table 8 brainsci-13-00147-t008:** Confusion matrix obtained by RNN on the Zain dataset.

RNN_Zain	Precision	Recall	F1-Score
Negative	0.98	1.00	0.99
Positive	1.00	0.98	0.99
Accuracy	-	-	0.99
Macro average	0.99	0.99	0.99
Weighted average	0.99	0.99	0.99

**Table 9 brainsci-13-00147-t009:** Validation loss results obtained by CNN, RNN, and AraBERT.

Model\Dataset	Zain	Mobily	STC
RNN	0.0412	0.2141	0.004
CNN	0.0504	0.2895	0.0099
AraBERT	0.04	0.10	0.03

**Table 10 brainsci-13-00147-t010:** Validation accuracy obtained by CNN, RNN, and AraBERT.

Model\Dataset	Zain	Mobily	STC	Average
RNN	0.9588	0.7859	0.9960	0.9135
CNN	0.9496	0.7105	0.9901	0.8834
AraBERT	0.96	0.90	0.97	0.9433

## Data Availability

The datasets has been taken from this link https://peerj.com/articles/cs-510/.

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
