# Peer review of "Using the AraBERT Model for Customer Satisfaction Classification of Telecom Sectors in Saudi Arabia"

_brainsci, 2023, doi:10.3390/brainsci13010147_

Round 1

Reviewer 1 Report

This paper presents sentiment analysis-based customer feedback prediction based on Twitter Arabic datasets of telecommunications companies in Saudi Arabia, where CNN, RNN and araBert based methods were implememted and compared.

Strength: this paper summarizes a set of DL-based methods towards sentiment analysis. The data preprocess is clear including stemming, removing irrelevant data and text segmentation. On the other hand, this paper first adopted araBert to predict and classify the ttelecommunication sector’s customer satisfaction, which is never been done before. Finally, by comparing CNN model and RNN model, using arabert can avoid overfitting and get better accuracy.

Some weakness:

1 it is better to give concrete examples to show the merits of arabert

2 more explanation on arabert is necessary, for example, the dataset for training arabert.

Reviewer 2 Report

Medium level article

Author Response

Thank you for your comments.

Reviewer 3 Report

This manuscript proposed a novel deep learning-based method for customer satisfactory classification of telecom sectors in Saudi Arabia, where Arabic Bidirectional Encoder Representations from Transformers (AraBERT) model was developed for the task of interest. To validate the performance and superiority of the proposed method, a comparative study was conducted by comparing the proposed one with Convolutional Neural Network (CNN) and Recurrent Neural Network (RNN). The comparison results demonstrate that the proposed AraBERT performed best among three deep learning models. Overall, the topic of this research is interesting, and the manuscript was well organised and written. I suggest that it can be accepted for publication in Brian Sciences, if the authors can well address the following comments.

1.       The main innovation and contributions of this manuscript should be clearly clarified in both abstract and introduction.

2.       Broaden and update literature review on fundamentals of deep learning or CNN/RNN and their practical applications. E.g. Torsional capacity evaluation of RC beams using an improved bird swarm algorithm optimised 2D convolutional neural network; Vision-based concrete crack detection using a hybrid framework considering noise effect.

3.       It is well known that the performance of deep learning model is mainly dependent on the setting of model hyperparameters. How did the author optimise them in this research to achieve the best classification performance?

4.       It will be better if the authors can use a different way to demonstrate classification performance of models. E.g. use a graph to describe the confusion matrix.

5.       The running time for training and test of developed model should also be considered as a metric.

6.       How about the robustness of the proposed model against noise effect?

7.       More future research should be included in conclusion part.
